



# The DataHawk2 Uncrewed Aircraft System for Atmospheric Research

Jonathan Hamilton[1,2], Gijs de Boer[1,2,3], Abhiram Doddi[4], Dale A. Lawrence[4]

[1] Cooperative Institute for Research in Environmental Sciences, University of Colorado Boulder, Boulder, Colorado, 80309, USA
[2] Physical Sciences Laboratory, NOAA, Boulder, Colorado, 80305, USA
[3] Integrated Remote and In Situ Sensing, University of Colorado Boulder, Boulder, Colorado, 80303, USA
[4] Aerospace Engineering Sciences, University of Colorado Boulder, Boulder, Colorado, 80303, USA

*Correspondence to*: Dale A. Lawrence (dale.lawrence@colorado.edu)

**Abstract.**

The DataHawk2 (DH2) is a small, fixed wing uncrewed aircraft system, or UAS, developed at the University of Colorado (CU) primarily for taking detailed thermodynamic measurements of the atmospheric boundary layer. The DH2 weighs 1.7 kg and has a wingspan of 1.3 m, with a flight endurance of approximately 60 minutes, depending on configuration. In the DH2's most modern form, the aircraft carries a Vaisala RSS-421 sensor for pressure, temperature, and relative humidity measurements, two CU-developed infrared temperature sensors, and a CU-developed finewire array, in addition to sensors required to support autopilot function (pitot tube with pressure sensor, GPS receiver, inertial measurement unit), from which wind speed and direction can also be estimated. This paper presents a description of the DH2, including information on design and development work, and puts the DH2 into context with respect to other contemporary UAS. Data from recent field work (MOSAiC, the Multidisciplinary drifting Observatory for the Study of Arctic Climate) is presented and compared with radiosondes deployed during that campaign to provide an overview of sensor and system performance. These data show good agreement across pressure, temperature, and relative humidity, as well as wind speed and direction. Additional examples of measurements provided by the DH2 are given from a variety of previous campaigns in locations ranging from the continental United States to Japan and northern Alaska. Finally, a look toward future system improvements and upcoming research campaign participation is given.

## 1 Introduction

The lower atmosphere plays critical roles in regulating weather and climate, and thereby has direct impacts on the daily lives of most of Earth's inhabitants. The interactions between the atmosphere and underlying surface result in the generation of turbulence and atmospheric mixing, govern heat transfer into and out of the surface of the earth, support development of clouds, fog, and precipitation, drive the lifecycles of hurricanes, thunderstorms and other forms of extreme weather, and drive air quality related to both anthropogenic and natural sources of atmospheric particles and gases. The air extending between the surface of the Earth 10–3000 m overhead typically includes a



surface-driven mixed layer and the planetary or atmospheric boundary layer (ABL). These layers generally feature well-developed mixing of atmospheric properties resulting from both surface-induced drag, as well as from the vertical transport of quantities through convection resulting from either the heating of the Earth's surface or other stratification within the atmosphere (e.g., longwave cooling at the top of stratiform cloud layers).


Given the influence of this layer on understanding the physical and chemical processes that drive our weather and help us to understand future climate states, and the importance of characterizing these processes and being able to correctly simulate them in support of weather prediction and climate projection, it is hardly surprising that numerous field campaigns are conducted every year to study in detail various elements of the ABL. Such campaigns generally

feature a focused observing effort that aims to capture new data on specific processes that are deemed to be particularly important, with such observing efforts generally coupled with years of analysis and model development and improvement work to help translate such knowledge into improved prediction of weather and climate. In support of such efforts, a variety of observing platforms have been developed and deployed. These include a variety of remote sensing systems, such as lidar and radar systems to better understand the thermodynamic and kinematic structure of

the lower atmosphere (e.g., Wilczak et al., 1996; Engelbart et al., 2007; Shupe et al., 2008). Additionally, this could include surface-based in situ sensing systems mounted on towers or mobile platforms (e.g., Li et al., 2010, Wolfe and Lataitis, 2018) to collect high-resolution, detailed information on the state of the atmosphere in a given location.

In addition to the remotely-sensed and surface-based observations, in situ observations have also been collected at

altitude leveraging a variety of platforms, including research aircraft, radiosonde and dropsonde systems, tethered balloon systems, and uncrewed aircraft systems (UAS)[1]. While all these platforms have contributed significantly to our understanding of the lower atmosphere, each have independent strengths and weaknesses. Remotely-sensed observations often provide extended time series of data due to the ability of these systems to operate continuously. Additionally, they can provide volumetric information leveraging the scanning capabilities of some systems. However,

the measurement principles applied can come with significant uncertainty, in part related to the composition of the atmosphere at any given time. For example, many radars operate at frequencies optimized to collect information on clouds and precipitation. However, this makes it challenging to collect data in areas of clear air, where no hydrometeors are present. Similarly, lidar systems use shorter wavelengths and can see clear air, assuming there are enough particles in the atmosphere to support backscatter towards the sensor system. However, lidars have an opposite

problem to radars in that they are readily attenuated by cloud cover, limiting their range in cloudy or precipitating conditions. Surface sensing systems typically also offer the ability to collect extensive time series, but suffer, with some very limited exceptions, from an inability to extend beyond a few meters from the surface of the Earth. Radiosondes (launched from the ground) and dropsondes (dropped from aircraft) can cover a larger range of altitudes, but only provide a single profile through the atmospheric column, thereby failing to capture details on the spatio-

temporal variability of atmospheric state at a given level. Research and commercial aircraft provide an ability to inform us on spatial and temporal variability, though commercial platforms tend to spend very little time in the ABL. Research

---

[1] Also known as drones, remotely-piloted aircraft, or unmanned aircraft systems



aircraft can cover these lower altitudes but are also limited due to operating expense and considerations of pilot and crew safety in hazardous environments, such as those connected to severe weather or remote operations. Finally, tethered balloon systems offer a nice ability to sample throughout the lowest 1–2 km of the atmosphere but are typically operated from a single location in space, making it difficult to observe location-dependent gradients such as those which may be present in a coastal zone. Operation of these tethered balloon systems can also be very limited by adverse weather conditions, particularly in relation to elevated wind speeds.

UAS fill a unique niche in measuring the atmosphere, adding perspectives that are not obtainable and/or safe to obtain with other in situ sensing methods. They can provide observations in a wide range of atmospheric conditions, some of which prove challenging for remote sensing-based methods. UAS can provide observations at altitudes from single meters above the surface all the way up through the upper troposphere, a much greater range than surface-based sensing allows. They can provide greater temporal and vertical resolution than radio/dropsondes and can fly in more "risky" situations than manned aircraft (e.g., closer to the ground, or in severe weather). Additionally, they provide greater horizontal resolution than tethered balloons, along with the ability to operate in higher wind conditions.

UAS have been used to investigate the boundary layer and lower troposphere dating back as far as 1970 (Hill et al., 1970). Recently, the advent of small, advanced, low-cost avionics has enabled the development of many UAS for atmospheric research purposes and has enabled collaborative use of different types of UAS during a single research campaign. An example of this collaboration is the LAPSE-RATE campaign in the San Luis Valley of Colorado (de Boer et al., 2020). The ease of access to avionics has also enabled UAS to be tailored for the investigation of specific phenomena; for instance, wind turbine wakes (Båserud et al., 2016), or vertical wind velocity measurements for an aerosol-cloud interaction study (Calmer et al., 2018). UAS can also augment more traditional types of instrumentation present, such as during BLLAST (Reuder et al., 2016). Due to their rugged nature and relative expendability, UAS have become a valuable tool for research in extreme environments, where they can help evaluate models and augment data from other sources as was done on the Ross Ice Shelf in Antarctica (Wille et al., 2017). Most recently, UAS have blended the latter two uses on MOSAiC, an icebreaker-based, multi-disciplinary Arctic research campaign, where they helped extend the reach of more traditional measurement techniques (de Boer et al., 2022b).

Both rotary wing and fixed wing UAS have been used at many of the campaigns mentioned above, and each platform has its individual merits. Rotary wing UAS are easy to operate from a small area, require less pilot training, and are more easily suited to very low (under 10 m AGL) flight regimes due to their ability to maintain altitude with no forward velocity. However, they generally lack endurance and are limited in their ability to measure 3D winds (Prudden et al., 2018). For example, the CopterSonde system developed by the University of Oklahoma has an 18.5-minute flight time (Segales et al., 2020), where a fixed wing aircraft in the same weight class, the DataHawk2 (abbreviated as DH2), can fly for 60 minutes. Fixed wing UAS can carry larger payloads over a longer distance and can more easily measure 3D winds as opposed to rotary wing UAS but require additional pilot training and a larger operating area.


Using the categories given in Elston et al. (2015) and the groups given in a publication assembled by the Department of Defense (Army UAS CoE Staff, 2010) as rough guides, fixed wing UAS can be classified based on physical dimensions and performance. Very large UAS (> 600 kg gross takeoff weight), such as the NASA Global Hawk (Naftel, 2009), fall outside the scope of this introduction, and the budgets of institutional operators. One step below these very large UAS are a variety of UAS exceeding a minimum takeoff weight of 25 kg and going up to 600 kg. These aircraft require extensive operator and maintenance training due to their complexity and cost and are generally supported by larger programs. Examples of such platforms include those previously operated by US Department of Energy (DOE) Atmospheric Radiation Measurement (ARM) Unmanned Aerospace Vehicle Program (Stephens, 2000), the current DOE ARM ArcticShark, and the University of Alaska Fairbanks SeaHunter UAS. These aircraft have benefits in terms of endurance and payload capability but are inaccessible to many potential research users due to their high cost and are ill-suited to high-risk situations that could cause the loss of an aircraft. Given the cost and complexity of these systems, here we focus our attention on small UAS (sUAS), those able to fly at a gross weight under 55 lbs (the maximum for US Federal Aviation Administration (FAA) Part 107 operation), or ~25 kg. Over the past decades, there have been several aircraft developed and deployed that weigh near the limit for sUAS. Some examples include the University of Colorado Pilatus UAS (de Boer et al., 2016) and the L3/Harris FVR-55 aircraft currently being developed for use by NOAA. Below these systems, new classes of sUAS have seen substantial campaign use over the past decade. These include aircraft made of rigid composite materials (e.g. carbon fiber) and more resilient materials, such as foam with an outer film-type skin. An example of a rigid aircraft is the University of Colorado (CU) Tempest, a 6.4 kg carbon fiber aircraft with a wingspan of 3.2 m, while the more resilient side of the spectrum could be filled by the CU RAAVEN (e.g., de Boer et al., 2022a), a 7 kg aircraft constructed primarily of foam with a wingspan of 2.3 m. These are large enough to carry multiple types of instrumentation and can be un-packed and set up relatively quickly, allowing for rapid deployments targeting rapidly-evolving weather situations (e.g., convective storms, mesoscale fronts). These sUAS cost substantially less than the large or very large categories, but the cost per aircraft instrumented is often still in the multiple thousands to low tens of thousands (USD), with the cost increasing dramatically with aircraft size.

In recent years, the development and adaptation of smaller fixed wing sUAS has significantly lowered the cost of performing atmospheric research with a fixed wing unmanned aircraft. One of the most popular fixed wing small sUAS designed specifically for atmospheric research has been the Small Unmanned Meteorological Observer or SUMO (Reuder et al., 2009). In its original form, this aircraft had a wingspan of 0.8 m, weighed ~0.6 kg, and could fly for up to 30 minutes. It was instrumented to measure pressure, temperature, and relative humidity. Since its initial development, it has been used for a wide variety of atmospheric research campaigns, with geographical locations ranging from Spitsbergen (Reuder et al., 2009), to Antarctica (Cassano, 2014), to more moderate latitudes such as Lannemezan, France during the BLLAST campaign (Reuder et al., 2016). Additionally, it was equipped with a miniature multi-hole probe (MHP) for turbulent flow measurements for select flights during BLLAST (Båserud et al., 2016). Aircraft in this size class bring the benefit of a very low cost per aircraft and the ability to ship multiple aircraft in a small space. They are still able to carry multiple sensors and can be operated in very remote locations.





The DataHawk2 (DH2) UAS falls into the same smaller sUAS class as the SUMO, with a wingspan of 1.3 m, weight of 1.7 kg, and airspeed ranging from 10-20 m/s, and has also been used in field deployments spanning a variety of geographical regimes, including Japan during ShUREX (Kantha et al., 2017), Utah during the IDEAL campaign (Doddi, et al., 2021), Colorado during the LAPSE-RATE campaign (de Boer et al., 2020; 2021), Northern Alaska during the POPEYE and ERASMUS campaigns (de Boer et al, 2018; 2019), and on Legs 3 and 4 of MOSAiC in the high Arctic (de Boer et al., 2022b). Like the SUMO, the DH2 is able to carry a variety of instruments tailored to investigating specific phenomena, but has a long flight time (approximately 60 min) for an aircraft of its size and is exceptionally durable. Additionally, one of the more unique sensors developed for the DH2 is a finewire array that provides measurements of airspeed and temperature at very high frequency, enabling it to measure smaller turbulent scales than a multi-hole probe equipped aircraft.

The DataHawk2 is custom-designed and constructed at the University of Colorado Boulder. It can collapse into a very small volume, which enables the easy transport of multiple DH2 aircraft to remote locations, and, together with its low cost per aircraft (about 1000 USD for the airframe and avionics), this makes it well suited to extreme operating conditions that would prevent deployment of more costly aircraft. The DH2 relies on a custom-developed autopilot and data-logging system that offers significant opportunity for customization and modification to support specific sampling objectives. An example of such customization was the addition of a dual-GPS based heading solution for high-Arctic operations near the magnetic north pole during the MOSAiC project (de Boer et al., 2022b). The costs for a fully equipped DH2 exceed the airframe cost (sensors add about 1000 USD), and customizations such as the dual-GPS system can also add cost (the most recent version of this system is about 500 USD). Additionally, it should be noted that the costs mentioned here are specific to an educational environment, where building DH2s provides opportunities for students to gain experience while constructing aircraft; these costs are not representative of the total system cost if one were to produce the aircraft commercially.

This paper provides a detailed overview of the DataHawk2's unique capabilities for ABL measurements, including its airframe, avionics, and scientific payload. In addition, we provide a detailed evaluation of sensor performance and the comparison of observations from the DH2 to those from other surface- and air-based sensors (e.g., radiosondes) that were deployed alongside the UAS during recent field campaigns. Beyond this, brief example usage cases from recent field studies are provided, providing insight into how the platform has been deployed in the past. Lastly, a look to possible future uses and improvements will be given.





## 2 DataHawk2 Description

### 2.1 Airframe

The DataHawk2 embodies many improvements over the original DataHawk sUAS (Lawrence and Balsley, 2013) based on hundreds of flight hours conducted over a variety of geographical and meteorological regimes. Similar to the SUMO, the original DataHawk design used a commercial molded expanded polyolefin (EPO) foam airframe (the Hobby Zone Stryker) which was attractive due to the very low initial cost. However, field experience revealed

shortcomings in ruggedness that led to frequent repairs and a short life span, increasing operational and maintenance costs. Although much of this damage could have been avoided by selecting a smooth, forgiving landing area and by using a skilled radio control (RC) pilot, these luxuries were often limited in the field campaigns of interest. The airframe was also found to be difficult to operate in high winds. This was most critical during launch and landing, making operation in windy conditions difficult and thereby restricting the conditions that can be sampled. Beyond

flight operations, the one-piece molded airframe occupied a large volume for its wingspan, requiring a correspondingly large container for shipping. This made it expensive to bring more than a few airframes to any field campaign, undercutting the advantages of a low-cost aircraft for redundancy and maintaining availability throughout a lengthy campaign. Finally, another limiting quality of the original airframe was that, as with many off-the-shelf products, the long-term supply was unpredictable.

Based on these experiences, the DH2 airframe (see Fig. 1) was designed with the following characteristics and procedures to reduce overall cost and improve field operability:

- Gust-insensitive aerodynamic design with no wing sweep or dihedral, and a vertically symmetric tail to eliminate the roll moment due to sideslip, resulting in neutral lateral stability and a natural tendency to

weathervane into a gust, rather than roll away from it.
- Elimination of protruding fuselage or empennage that can be broken off easily, resulting in a compact "flying wing" design with a strong, wide body and a blunt nose.
- Eight-piece segmented design, allowing removal of wings, fins, and motor mount so the entire airframe can be packed in an efficient 9 cm by 31 cm by 67 cm rectangular volume ($< .02$ m$^3$), enabling 5 aircraft and

associated support equipment to be shipped in a single 88 cm by 67 cm by 41 cm case (0.25 m$^3$ volume).
- Use of tougher, more elastic expanded polypropylene (EPP) foam that returns to its shape after impact.
- Custom cutting of foam shapes on a commercial hot wire foam cutter, enabling a continuous supply of parts. This leaves open foam cells on the cut surface that must be covered by a thin lamination/glue combination to provide a smooth, waterproof skin.

- Design of a custom lightweight but high-strength aluminum motor mount that flexes rather than breaks during hard landings.


- Incorporation of internal carbon fiber spars in the body and wings for stiffness but connected by flexures that allow the wings to bend forward rather than break spars on hard landings.
- Use of hollow, triangular fiberglass trailing edges and control surfaces that flex on impact rather than permanently "crease".
- Use of fiberglass fiber tape at key locations on the leading edges and on the body and wings to connect and stiffen the structure, providing overall toughness and strength with very little weight.
- Direct-drive servo connections to control surfaces, eliminating exposed control rods/horns that can be damaged.

The resulting aircraft is very rugged and is rarely damaged from hard landings in rugged terrain. Typically, accumulation of abuse results in a loosening of the exterior tensioning tape, but this is easily replaced with new tape to restore the rigidity of the airframe. If repairs are needed, spars and foam sections can be easily cut out and their replacements glued in place.

## 2.2 Avionics

There are a variety of avionics on board modern sUAS. Servos for control surfaces and speed controllers for the propeller motor have advanced rapidly and now contain programmable microprocessors to set a variety of operating modes and safety limits. These are relatively independent of other avionics, and there are a wide variety of off-the-shelf options to choose from. Similarly, manual flight control through a RC radio link has several sophisticated commercial options. More complicated is the choice of autopilot avionics and associated signal conditioning and data handling for on-board scientific sensors.

When the original DataHawk was developed, there were no suitably-small and low-cost autopilot systems available, so one was developed in-house as part of a Ph.D. thesis (Pisano, 2009). At the time of the DataHawk re-design, many of the original autopilot avionics components had become obsolete, and a re-designed custom autopilot was developed. This process was undertaken with two primary considerations: 1) Developing hardware architecture to keep up with the constant innovation (and obsoleting) of key autopilot components (e.g., inertial sensors and GPS receivers), and 2) providing a software foundation to support continuous advances in measurement and operational techniques required by the scientific community.

With these considerations in mind, the DataHawk2 took a modular approach, separating the functions of the microprocessor, power conditioning, flexible connection to peripherals, and multiplexing between autopilot and RC manual control of the control surfaces and propulsion between multiple boards. The processor was upgraded to a 32-bit ARM microcontroller clocked at 180MHz, with a floating-point co-processor. This enables updates to many different components to be localized to that corresponding board, without requiring wholesale changes elsewhere. It also enables components to be located optimally on the airframe, helping to reduce interference of motor currents on



the magnetometer, and reducing multipath reflections on the GPS antenna. Figure 1 shows where the components of the autopilot and sensors are in the airframe.

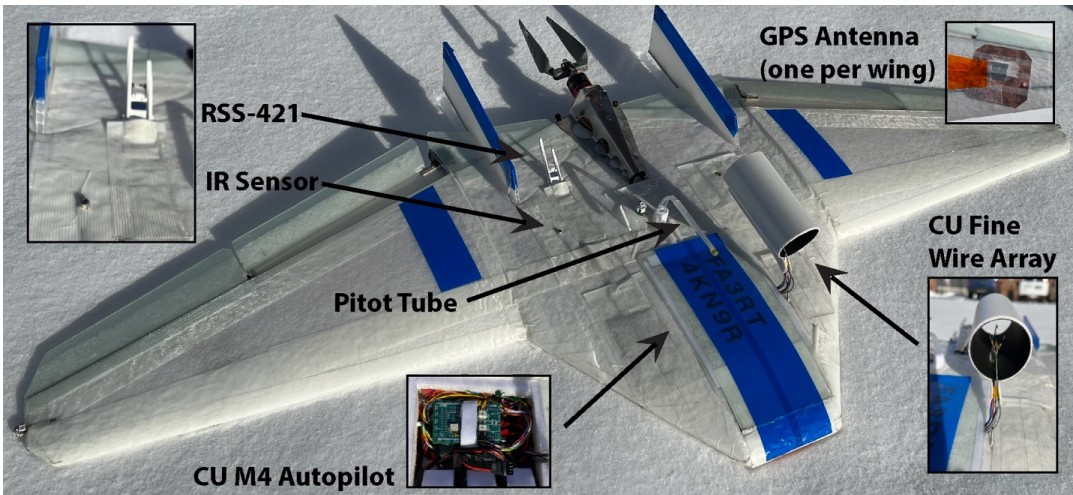


**Figure 1:** The DataHawk2 sUAS. Detailed images show close ups of individual sensing systems and point out where these are located.

Flight software presented a more difficult trade-off. While commercially-available autopilot software is extensively tested, it can be daunting to modify such a large code base without intimate knowledge of its architecture. Simultaneously, building a custom code base has the advantage of complete version control, and comes with intimate knowledge that enables modifications and customization. As a result, the choice was made to develop custom software for the DH2. The autopilot processor also handles sensor data, storing it at native rates on a microSD card, and sending 260 a subset of these data to the ground station at lower rates for real time sensor monitoring during flight.

The autopilot takes a vector field control approach (Lawrence et al., 2008), causing the aircraft to be attracted laterally to specified circles or lines. Vertically, the aircraft tracks specified rates of climb/descent, bounded by specified ceiling and floor parameters. For example, a repeating helical vertical profile is provided by selecting a horizontal circle center 265 location and radius, climb/descent rates, and ceiling and floor altitudes. Lower-level control loops track compass heading, airspeed, elevation angle and bank angle to track the vector field, with active compensation for current wind conditions that modifies the commanded compass heading to produce the desired GPS course heading. Airspeed is sensed with a miniature pitot-static tube. Heading, elevation, and bank angles (aircraft attitudes) are estimated by fusing 3-axis accelerometer, gyroscope, magnetometer, and "moving-base" differential GPS in a simplified Kalman 270 filter algorithm that runs at 100Hz. The latter two measurements enable reliable attitude estimation in high-latitude locations where the local magnetic field vector is nearly co-linear with the gravity vector.


Experience operating in the restricted airspace R-2204 at Oliktok Point Alaska, in proximity to wind profiling radars and an Air Force early warning radar, prompted several avionics modifications to avoid anomalies in flight control.

Early campaigns there experienced a range of intermittent anomalies from sensor glitches to GPS mistracking to catastrophic processor execution halt. In response, a hardware multiplexing scheme was developed to allow manual control of aircraft elevons and propulsion to override the autopilot by direct RC command, independent of the state of autopilot processor execution. This provides a fail-safe backup in case of autopilot failure for any reason. Also, an in-flight processor reset capability was added, where the entire state of the flight control system is continuously stored

in non-volatile memory so that the processor can be reset, then the previous flight state can be restored for a smooth continuation of the flight. These resets can be automatically generated by detections of peripheral sensor anomalies, or watch-dog time-out if the processor execution stops. Resets can also be manually generated from the ground station. Software protections include detection of a plugged pitot-static airspeed sensor (e.g. from rain drops or icing), and detection of erroneous GPS tracking to prevent upsetting of the state estimation and control system. Mitigations of

such "off-nominal" operation challenges further enhance the ruggedness of the system over off-the-shelf options, and these mitigations are enabled by the custom hardware and software development used in the DH2.

### 2.3 Scientific Payload

The DataHawk2 is primarily equipped to make detailed measurements of the thermodynamic structure of the

atmosphere. To support such measurements, the system has carried a variety of sensors throughout its history. The most recent version of the DH2 has included a Vaisala RSS-421 pressure, temperature, and humidity (PTH) sensor suite embedded in the airframe foam, with the sensors extended into the streamflow that passes over the aircraft. The RSS-421 is similar to Vaisala sensors that are used commonly as radiosondes (RS-41) and identical to the sensor suite integrated into the Vaisala dropsonde system (RD-41). The platinum resistive temperature sensor on the RSS-421

offers 0.01 °C resolution and measurement repeatability of 0.1 °C with a response time of around 0.5 s at typical airspeeds. The capacitive silicon pressure sensor has a resolution of 0.01 hPa with a repeatability of 0.4 hPa. Finally, the thin-film capacitive relative humidity (RH) sensor includes active sensor temperature monitoring and correction, offering a resolution of 0.1 % RH, a repeatability of 2 % RH and a temperature-dependent response time that ranges from approximately 0.3 s (at 20 °C) to 10 s (at -40 °C). Previous versions of the aircraft also employed an iMET-1

radiosonde sensor system developed by interMet Systems, though this sensor is not currently used in the DH2.

In addition to the Vaisala sensor system, the DH2 carries a custom finewire array that was developed at the University of Colorado. This consists of 5 μm diameter platinum sensor wires, one operated as a coldwire thermometer and one as a hotwire anemometer using a custom electronics board. The array also includes a Sensiron SHT-85 temperature

and humidity sensor. This array was modified for use at high latitudes following a test campaign on the Svalbard Archipelago. Initially, the shroud was constructed of aluminum, but this caused multipath issues with the differential GNSS antennas integrated on the wings for high-latitude operations. This issue is exacerbated by the relatively low position on the horizon of the GNSS satellites at high latitudes. The design was modified to have a foam-covered plastic shroud which mitigated these GPS issues while still retaining the insolation-shielding properties of the original



design. The new shroud design is also a result of detailed wind tunnel studies characterizing the secondary turbulence generation of finewire protections against contact with airborne particles. Despite the small scale of these protective obstructions, it was found that the additional turbulence generated has enough cascading energy at larger scales to affect the parameterization of geophysical turbulence. Thus, many of the protections used previously (small shields up-stream of the wires, rear-facing wires, etc.) were removed and the shroud diameter was increased from 1 cm to 3

cm. Comparisons between free-stream finewire placement with those inside the new shroud in DH2 fight tests showed negligible impact on the portion of the inertial sub-range used for turbulence parameterization. Because of this new design, finewire breakage does occasionally occur in flight if precipitation is present, and sometimes upon landing where snow or vegetation fragments can be kicked up, but generally the finewires are robust enough to withstand operation in rugged terrain. The finewires themselves are produced in batches using the Wollaston wire technique in

the laboratory at CU. At a cost of < \$5 each, wire breakage is not a cost driver, and wires are easily replaced in the field.

The voltage signals from the finewire electronics are converted to fluctuations in relative wind velocity and temperature through post-flight calibration. Spectral analysis can then be used to fit a Kolmogorov inertial sub-range

model to the power spectral density as a function of frequency, and mean velocity is used to convert frequency to wavenumber. These spectral fits can then be converted to infer information on turbulent characteristics of the atmosphere, such as kinetic energy dissipation rate $\varepsilon$ (from the hotwire) and temperature structure parameter $C_T^2$ (from the coldwire) (e.g. Frehlich et al., 2003).

Contemporary UAS like the MMAV (van den Kroonenberg et al., 2008), MASC (Wildmann et al., 2014), BLUECAT (Witte et al., 2016), SUMO (Båserud et al., 2016), Skywalker X6 (Calmer et al., 2018), and OVLI-TA (Alaoui-Sosse et al., 2019) have shown proof of concept for turbulent wind measurement using high-cadence multi-hole pressure probes and finewire (Witte et al., 2016) sensors typically for measurements in the atmospheric boundary layer. However, the authors report that due to the elevated noise floor of the multi-hole pressure sensors the effective

bandwidth of the sensors is limited to 40-100 Hz. This inhibits most UAS from measuring small-scale, weak turbulence structures typically found in the free atmosphere. The DH2 is equipped with a custom finewire anemometer and thermometer that sense airspeed and temperature at a cadence of 800 Hz. The low white noise floor of the custom finewire turbulence sensors on the DH2 enables the DH2 to measure turbulence in scales as small as ~0.0375 m (15 [m s$^{-1}$]/400 [Hz]; assuming 15 m s$^{-1}$ nominal flight speed).


Finally, the DH2 carries a pair of infrared temperature sensors. These sensors offer information on surface heterogeneity below the aircraft, as well as cloud cover above. Such information can be useful when attempting to associate changes in atmospheric conditions with surface features such as coastal boundaries, leads in sea ice, lakes or ponds, or vegetation coverage. The sensors are based on a custom design that utilizes the 10TP583T thermopile, in

combination with amplification and compensation from an integral case-temperature thermistor, to provide an approximate optical temperature of the area in the sensor's approximately 90-degree conical field of view. One sensor





is mounted with a view above the aircraft, and one is mounted with a view below, providing temperature variation information from the sky and from the surface (see Fig. 2). The sensor time constant of 15 ms and the 100 Hz sampling enable fast variations of ground features to be captured at the typical flight speeds of the aircraft.


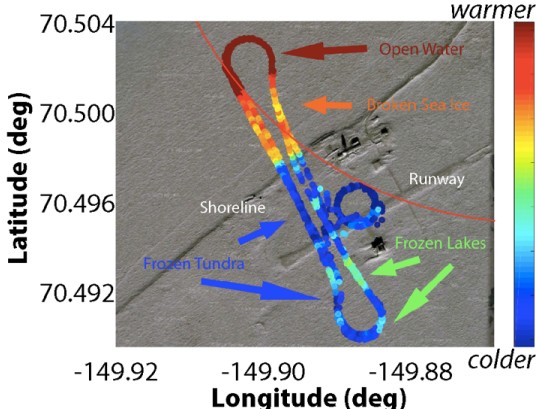

**Figure 2:** An example of data from the downward-looking IR sensor from a fall flight at Oliktok Point, Alaska during freeze-up of the surface. The thermopile voltage plotted here is a proxy for the temperature of the target. The background image is © GoogleMaps, as downloaded using their API in 2014.


Data is logged by the autopilot to an integrated microSD card at 800 Hz for the hotwire and coldwire signals, at 5 Hz for the GPS signals and the RSS-421 signals, and at 100 Hz for the rest of the measurements. SD card write failures or in-flight autopilot resets can cause these signals to become unsynchronized, so all signals are time-aligned in post flight processing to GPS time within 200 ms.


## 3 Sensor Performance and Evaluation

A few factors may impact the uncertainty values given in the RD-41 dropsonde datasheet for the RSS-421 and discussed in Sect. 2.2, as the RD-41 is designed as a one-time use sensor. Vaisala includes an option to regenerate the

humidity sensor through a heating cycle to avoid the impacts of aging on the sensor accuracy. Under standard operating procedure, this process is conducted at least daily during DataHawk2 field campaigns. However, on MOSAiC, the RSS-421 sensors were failing irrecoverably frequently after undergoing this process, so, given the limited number of sensors on board and the inability to get more, the decision was made to generally forego this step. Second, the DH2 moves at a higher airspeed than the RD-41 descends close to the surface, which produces increased aspiration over its

sensors. Additionally, as mounted on the DH2, the RSS-421 does not have any solar shielding (similar to the RS-41 radiosonde), so solar effects could impact its measurements in certain cases, though flight data from MOSAiC does not show a significant dependence of temperature on solar angle (< 0.1 °C of variation present across all solar angles on the sensor). Lastly, flying in certain weather conditions can result in wetting (ex. the summer fog of MOSAiC Leg 4) or icing (ex. the cold winter of MOSAiC Leg 3) of the sensor, which could impact measurement quality.





To assess the field performance of the DH2 sensors, measurements from the aircraft (Jozef et al., 2021) are compared
to radiosonde-based observations obtained during the MOSAiC campaign (Maturilli et al., 2021). The data from each
DH2 sensor and the radiosonde parameters were averaged over 10 m altitude bins, starting with an altitude of 30 m
and extending to the top altitude of the DH2 flight. A paired t-test was chosen to investigate if there is a mean
difference between the radiosonde data and the data from various sensors on the DH2. Except for the derived standard
wind speed estimate, differences in the true mean difference between the radiosonde and DH2 observations were
found to be not zero (i.e., the null hypothesis of zero mean difference was rejected) at the 95 % significance level.
Therefore, a confidence interval was computed to determine a range for the actual difference between the radiosonde
and DH2 sensors given the same 95 % significance level. For each measurement, the standard deviation of the
difference between the DH2 and Radiosonde data bins are given in Table 1, along with the minimum and maximum
values showing the confidence interval. The data used in this comparison are limited to radiosonde data taken within
an hour of DH2 datapoint time, span the Arctic melt season (5 April – 26 July 2020) and are from flights conducted
in a variety of atmospheric conditions.

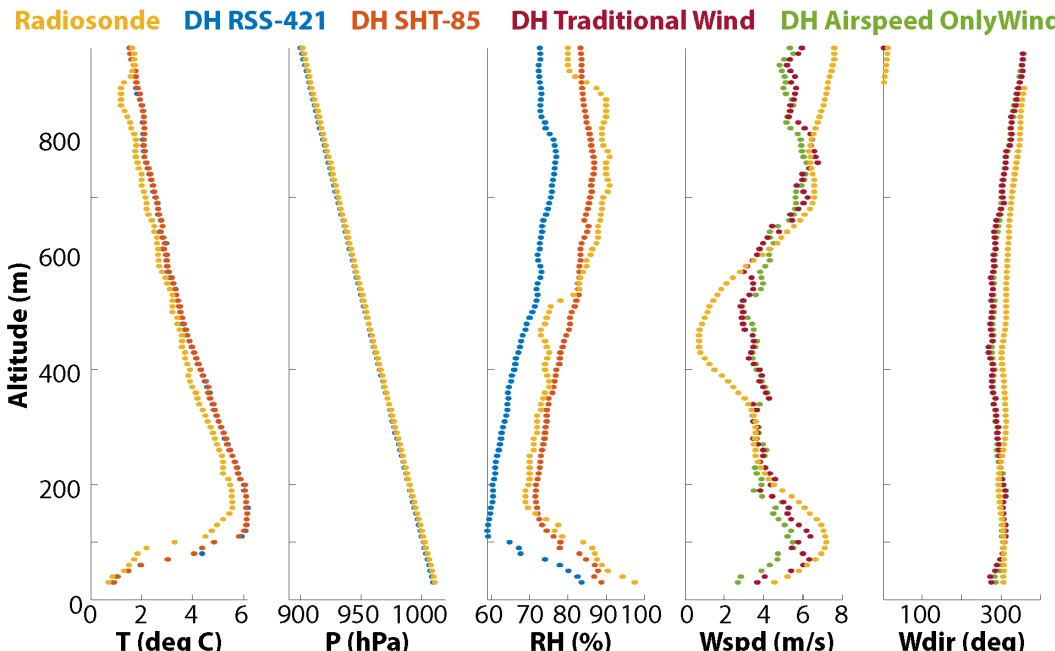


**Figure 3:** Example profiles from the DataHawk2 deployment for the MOSAiC experiment, in comparison with data
from a nearby radiosonde launch. Included are (from left to right) air temperature, air pressure, relative humidity, wind
speed and wind direction. Note that in this instance the RSS-421 was not conditioned prior to flight, resulting in a
significant low bias in relative humidity.





The data in Table 1 shows that the two temperature sensors present on the DH2 show similar errors relative to the radiosonde data, which exceed the repeatability value for the RSS-421 (0.1 °C) and SHT-85 accuracy of +/- 0.1 °C. This difference could be because of the effects mentioned previously (increased sensor aspiration, solar affects, wetting/icing of sensor), but it is also plausible that, given the difference in time (up to one hour) and lateral position (up to 2.3 km) of the measurements, a ~0.3 °C true difference in temperature is present. In the first panel of Fig. 3, a small difference (approximately the 0.3 °C shown in Table 1) can be seen over most of the profile, though this difference is slightly larger or smaller at certain points during the flight. Pressure shows good agreement between the RSS-421 and radiosonde, slightly exceeding the repeatability value (0.4 hPa) given in the RSS-421 datasheet. This similarity can be seen in the second panel of Fig. 3; little deviation between the two pressure sources can be seen. Greater differences both between the RSS-421 and SHT-85 and between each sensor and the radiosonde data are present in the relative humidity data, as seen in Table 1 and the third panel of Fig. 3. The SHT-85 differences exceed the stated accuracy (+/- 1.5 % RH) by a small amount, which seems reasonable given the difference in sensors and position/time of measurement. The RSS-421 has significantly more deviation from the radiosonde, however, well exceeding its repeatability value of 2 % RH. Outside of factors common with the other measurements, (differences in time and position of measurement), the larger difference in RH may be due to the lack of reconditioning, as mentioned in Sect. 2.2. The significant deviation of the RSS-421 from the other sensors is apparent in the example profile shown in the third panel of Fig. 3. Visualizations of the comparison data presented in Table 1 can be found in (de Boer et al., 2022b).

| Quantity (Sensor) | Standard Deviation | 95 % C.I. Minimum | 95 % C.I. Maximum |
|---|---|---|---|
| Temperature (RSS-421) | 0.430 °C | 0.299 °C | 0.340 °C |
| Temperature (SHT-85) | 0.469 °C | 0.298 °C | 0.342 °C |
| Pressure (RSS-421) | 0.623 hPa | -0.530 hPa | -0.472 hPa |
| Relative Humidity (RSS-421) ** | 5.686 % | -9.496 % ** | -8.929 % ** |
| Relative Humidity (SHT-85) | 5.950 % | -2.427 % | -1.834 % |
| Wind Speed (standard) | 1.58 m s$^{-1}$ | -0.12 m s$^{-1}$ | 0.04 m s$^{-1}$ |
| Wind Speed (hybrid) | 1.52 m s$^{-1}$ | -0.84 m s$^{-1}$ | -0.70 m s$^{-1}$ |
| Wind Direction (standard) | 20.31 deg | -3.02 deg | -1.09 deg |
| Wind Direction (hybrid) | 15.46 deg | -2.69 deg | -1.22 deg |

**Table 1.** DataHawk2 instrumentation or derived parameter difference from radiosonde data taken within an hour of a given data point. ** *Denotes that the RSS-421 relative humidity sensor was not often reconditioned during campaign, leading to the dry bias demonstrated here. This evaluation is not characteristic of the performance of this RH sensor, which has been demonstrated to provide accurate measurements of RH (e.g., de Boer et al., in prep).*

The high-rate finewire sensor measurements are calibrated against co-located (but slower) reference sensors in post flight data analysis. Voltages from the coldwire temperature sensor are calibrated against the reference temperature

from the SHT-85 sensor, located approximately 3 cm downstream inside the protective shroud on the file wire module.

As a result, the calibrated coldwire temperature inherits the uncertainty of the SHT-85. Due to the differing time-constants between the coldwire (about 0.5 ms) and the SHT-85 (about 2 s), these signals can be offset relative to each other when the ambient temperature is changing (e.g., in vertical profiling). Here, it is important to include both ascent and descent profiles as part of the calibration so that lag-induced offsets cancel each other out. The calibration improves with a larger range of temperature values (e.g., several °C or more), so that average signal excursion

dominates the turbulent fluctuations, reducing uncertainty in the calibration curve fit.

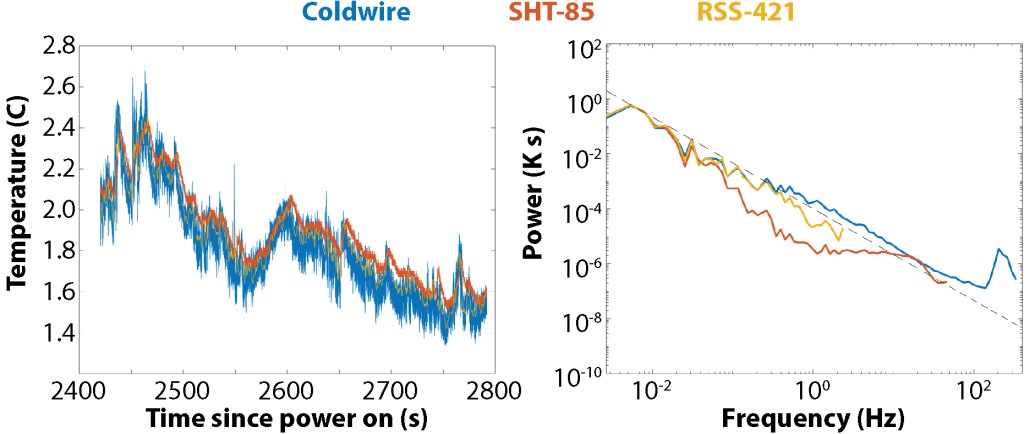

**Figure 4:** A timeseries (left) of temperatures measured during level flight by the DataHawk2 during the MOSAiC
campaign. Included are temperature values measured by the coldwire sensor (blue), RSS-421 sensor (yellow) and SHT-85 sensor (red). The power spectral density of the temperatures recorded by each sensor are included at right.

Figure 4 shows a comparison between the different temperature sensors carried by the DH2, based on data from an extended level flight leg in the Arctic boundary layer (Jozef et al., 2021). This example shows the different response

times of the individual sensors, and the reporting frequencies of each. The coldwire sensor, recorded at 800 Hz is able to record very fast fluctuations in temperature, though system noise becomes evident in this particular case around 100 Hz. The SHT-85 has a much slower response time, producing a roll-off in the spectrum above 0.1 Hz, and suffers from a relatively high level of signal quantization, that causes spectral flattening above 1 Hz. Finally, the RSS-421 is shown to have a response roll-off starting around 0.5-0.7 Hz due to the inherent time-constant of the sensor.


The hotwire sensor voltage is calibrated against pitot-static airspeed. Both sensors are located on the top of the aircraft, at about the same longitudinal and vertical positions with approximately 10 cm lateral offset. The measured pitot-static differential pressure $P$ is calibrated to first-order airspeed $v$ using the dynamic pressure formula

$$P = \frac{1}{2}\rho v^2 \qquad (1)$$






with an estimate of the local air density $\rho$ derived from altitude and the US standard atmosphere. This value is used for autopilot airspeed control and wind-aware guidance. A second-order correction to this pitot airspeed is conducted in post-flight analysis by comparing mean airspeed to extrema of GPS speeds during circular trajectory segments, adjusting pitot airspeed to lie midway between these GPS extrema. Only the turbulent fluctuations in airspeed are

sensed by the hotwire instrument, because an auto-zero process is active in flight to keep the hotwire voltage near the midpoint of the measurement range. Auto-zero adjustments are also recorded, so that re-calibration against pitot airspeed can be computed whenever the adjustments change (although this happens rarely during flight). Calibration for the hotwire data is calculated by comparing spectral data from the pitot airspeed and the hotwire voltage and adjusting the hotwire scale factor from V to m s$^{-1}$ to agree.


Calibrated velocity and temperature fluctuations, respectively, are used to parameterize turbulence intensity in terms of kinetic energy dissipation rate $\varepsilon$ and temperature structure parameter $C_T^2$. These parameters are computed via spectral processing in post flight analysis. The fast finewire response, high sample rate, and low electronics noise floor enable high-spatial resolution of these turbulence parameters. For example, if 1 s time records of the 800 Hz samples

are used, spectral analysis provides up to 2.6 decades of the inertial subrange to fit with the $f^{-5/3}$ characteristic Kolmorgorov cascade (Kolmorgorov, 1962), providing turbulence estimates averaged over a spatial interval of 15 m horizontally or 1 m vertically (assuming 15 m s$^{-1}$ airspeed and 1 m s$^{-1}$ ascent rate). Figure 5 shows a representative power spectral density of temperature fluctuations (blue line) and the fractional decade frequency bin averages (red dots), along with the Kolmorgorov cascade fit (black line) and the standard deviation of this fit (dashed lines). The

level of the fit is then converted to turbulence parameterization ($C_T^2$ in this case) according to (Frehlich et al., 2003). Details of this process, such as removal of spectral artifacts by choice of which bin averages (green dots) to use in fitting, are currently in preparation for publication, but similar methods can be found in (Luce et al., 2019).

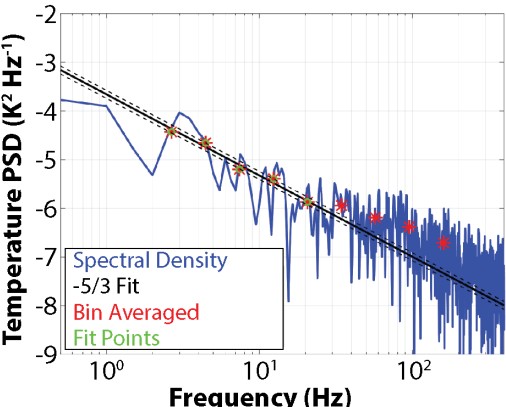

**Figure 5:** Spectral fitting process for estimating turbulence parameters ($C_T^2$ in this case) fits an inertial cascade model (black line) to the raw spectral data (blue line) by first averaging over fractional-decade frequency bins (red dots), using a subset (green dots) that are free of artifacts.



Finally, the DH2's IR sensors have been calibrated to provide only a relative measurement of infrared temperature. This can help one distinguish ground or sky features as mentioned in the previous section, but the CU IR sensors do not currently provide an accurate determination of optical temperature. Figure 2 provides an example of the perspectives offered by this sensor, leveraging data from a flight near Oliktok Point in Alaska, where the periods of flight over different surface features can be clearly seen on the IR sensor temperature colorized flight trajectory.
Additionally, it is important to note that the atmosphere is not entirely transparent to these sensors (3-15 μm spectral range), meaning that at altitude significantly different than that of the object whose temperature is being sensed, atmospheric contributions to the measured temperature may impact the readings.

Wind retrieval from a moving platform is a complex topic. Briefly, the DH2 has used both the "standard" approach, using attitude estimates to rotate body-frame relative wind measurements into Earth-frame coordinates to combine
with GPS velocity measurements in the wind triangle to derive wind estimates, and a "hybrid" approach that relies primarily on airspeed magnitude and GPS velocity, with only secondary use of attitude estimates. Both methods are susceptible to errors when the vehicle makes rapid maneuvers, e.g., during the downwind leg of tracking a circle in high winds, requiring judicious data excision of some intervals before applying the wind estimation algorithms.

The standard approach leverages the equations documented in the literature for wind estimation from aircraft. From the perspective of UAS, this technique is laid out clearly in Van Den Kroonenberg et al., 2008, where the zonal, meridional and vertical wind components are defined as:

$$
\begin{aligned}
u = v_{Ag} - |U_A|D^{-1}[(\cos\theta \sin\psi) &+ \tan\beta\,(\sin\phi\,\sin\Theta \sin\psi - \cos\phi \cos\psi) \\
&+ \tan\alpha\,(\cos\phi \sin\Theta \sin\psi + \sin\phi \cos\psi)]
\end{aligned}
\tag{2}
$$

$$
\begin{aligned}
v = u_{Ag} - |U_A|D^{-1}[(\cos\theta \cos\psi) &+ \tan\beta\,(\sin\phi\,\sin\Theta \cos\psi - \cos\phi \sin\psi) \\
&+ \tan\alpha\,(\cos\phi \sin\Theta \cos\psi + \sin\phi \sin\psi)]
\end{aligned}
\tag{3}
$$

$$
w = -w_{Ag} - |U_A|D^{-1}[(-1\sin\theta) + \tan\beta\,(\sin\phi\cos\theta) + \tan\alpha\,(\cos\phi\cos\theta)]
\tag{4}
$$

where $|U_A|$ is the true airspeed measured by the aircraft's air data system, $D$ is a function of the aircraft's angle of attack ($\alpha$) and sideslip ($\beta$) angles:

$$
D = \sqrt{1 + \tan^2\alpha + \tan^2\beta}
\tag{5}
$$

and $v_{Ag}, u_{Ag}$ and $w_{Ag}$ are the eastward, northward, and downward velocities of the aircraft relative to the ground, as measured by GPS, and $\theta, \psi$ and $\phi$ are the aircraft pitch, yaw, and roll angles, respectively, as measured by the inertial measurement unit. Unfortunately, the DH2 does not carry a sensor to measure angle of attack or sideslip, so for the purpose of estimating winds, those relative wind angles are assumed to always be constant and set to zero





(although the angle of attack could be set to any constant value, if desired, to account for an estimated average in-flight attack angle). This assumption makes the wind estimates insensitive to high-frequency lateral turbulent motions in the atmosphere, as the aircraft is not instantaneously weathervaning into the relative wind (estimated time constant is about 1 s), although the longitudinal turbulent components of the wind are not attenuated up to the 400 Hz Nyquist rate of the pitot-static sensor.

These calculations are also sensitive to misalignment between the axis of the aircraft's airspeed sensor and the UAS inertial measurement unit, and to differing time delay between the GPS and IMU derived variables. Additionally, they are very sensitive to biases in airspeed. To account for these issues, the measurements from the aircraft are put through an optimization routine that varies $|U_A|$, $\theta$ and $\psi$, with the latter two undergoing a full rotation to account for the impact of an adjustment to an individual axis to the values for the other two axes. To accomplish this, winds are 520 calculated for each combination of variables, and the variance in the wind estimate is minimized as the impact of angular offsets or incorrect $|U_A|$ is to create steps in the calculated winds as a function of heading, which increase variability in the derived winds.

Another approach to retrieving wind estimates has also been used on the DH2. This derives from the "airspeed only" approach (Lawrence and Balsley, 2013) that uses the geometry of the wind triangle along with measured GPS velocity 525 and pitot-static airspeed to constrain the horizontal wind vector to a circle at each time step. Wind estimates from the previous time step are projected onto this constraint circle along the direction of the current airframe compass heading, reducing the one-parameter family of solutions for the wind to a single solution at the current time step. This method reduces the sensitivity to variable delay in sensor data but can be biased by poor previous wind estimates. Methods to 530 counteract this error involve forward and backward in time wind estimate updates to cancel the directional bias. Both these wind retrieval methods are currently in development and validation by comparison with nearby radiosonde winds. However, the raw data from many of the previous campaigns is available for others to use in pursuing wind estimation approaches as well.

Both wind estimation techniques can be seen compared to the radiosonde wind estimates in Fig. 3, panels four (wind speed) and five (wind direction). For this example flight, the two wind estimation techniques agree quite well with one another and the radiosonde estimates, though they do deviate somewhat from the radiosonde wind velocity and direction estimates at certain points in the profile. Table 1 shows the agreement for the winds computed using radiosonde datapoints taken within one hour of DH2 datapoints, as fully described earlier in this section. From the 540 confidence intervals calculated from the MOSAiC radiosonde comparison, the DH2 standard approach shows very good agreement (< 0.12 m s$^{-1}$ difference) with the wind speed estimates from the radiosondes. The hybrid approach is within 1 m s$^{-1}$ of the radiosonde estimate but differs more than the standard approach. For wind direction, the standard approach has a wider confidence interval for the true difference from the radiosonde than the hybrid approach, but less difference between the two techniques is discernable here; both range from approximately 1 to 3 degrees offset





from the radiosonde. Given the difference in time and physical position between the DH2 and radiosondes, both wind estimation approaches seem reasonable for both wind speed and direction.

## 4 Previous Deployments and Scientific Use Cases

Since its redesign, the DataHawk2 has been deployed to a variety of locations. Through these deployments, the design

was further improved and refined, resulting in a robust and reliable platform capable of collecting in situ observations in the lower atmosphere over a variety of different climatological regimes. This section provides brief overviews of some of these deployments to provide further insight into platform capabilities and development.

One of the first locations that the DH2 was deployed to was arguably also one of the most challenging. Under funding

from the US Department of Energy, a team of University of Colorado researchers were deployed to Oliktok Point, Alaska (70.5103° N, 149.8600° W) to conduct a multi-week flight campaign and make detailed observations of the lower atmosphere. This field campaign, named Evaluation of Routine Atmospheric Sounding Measurements Using Unmanned Systems (ERASMUS), took place in August 2015. While the weather conditions during this time of year did not pose any significant issues, there were several obstacles that had to be overcome to successfully operate at this

facility. The primary obstacle was electromagnetic interference from the long-range radar facility operated by the US Air Force at this location, resulting in development of the in-flight reset capability for the autopilot that was discussed previously. Another change was an update to the autopilot software to use a combination of airspeed measured by the on-board Pitot-static probe and the ground velocity measured by the on-board GPS system to control throttle settings. These changes were implemented because of a clogged pitot event that occurred when the aircraft flew through clouds.

While there was significant concern about the influence of the high-latitude environment on both GPS and magnetometer performance, neither of these posed any challenges in the operation of the DH2 at this location. With the changes described above in place, the University of Colorado team was able to return in 2016 and conduct a successful flight campaign, collecting tens of hours of data between the surface and 1 km altitude, including extended low-altitude flight over the near-coastal Beaufort Sea (de Boer et al., 2018).


As part of ERASMUS, the DOE took ownership of a small fleet of DH2 aircraft. These systems were operated by the DOE ARM team both at Oliktok Point and in the continental United States for three years. Additional campaigns conducted by DOE using the DataHawk included the Inaugural Campaign for ARM Research using Unmanned Systems (ICARUS) and the Profiling at Oliktok Point to Enhance YOPP[2] Experiments (POPEYE) campaigns (de

Boer et al., 2018; de Boer et al. 2019). As with ERASMUS, these campaigns saw the DH2 conducting regular profiling between the surface and 1 km altitude over Oliktok Point, as well as collecting statistics at given altitudes throughout the atmospheric column. The latter flight mode included extended (30 minutes at a time) sampling at 20 m altitude above newly-forming sea ice in the near-coastal zone. In total, these campaigns resulted in 424 flights and 189.9 flight

---

[2] Year of Polar Prediction



hours. Additionally, these campaigns helped to demonstrate the platform as a viable high-latitude data collection
mechanism, setting the stage for future deployments (e.g., MOSAiC). The data collected as part of ERASMUS,
POPEYE and ICARUS continue to be leveraged for scientific investigations. As an example, some of the data from
ERASMUS are currently being used to evaluate small-scale turbulent structures in stable boundary layer conditions
(B. Butterworth, in prep).

The DH2 was also used extensively for the Instabilities, Dynamics, and Energetics Accompanying Layering (IDEAL)
campaign at Dugway Proving Ground (DPG), Utah, for a 23-day period in November, 2018 (Doddi et al., 2021). The
focus of this campaign was turbulence characterization in stratified flows, conducting nighttime observations within
and above the nocturnal boundary layer using sorties of up to 3 simultaneous DH2 flights. These flights were
conducted alongside continuous 900 MHz wind-profiling radar data and coordinated radiosonde releases from
NCAR's Integrated Sounding System (ISS) and data from DPG's distributed surface measurement system (10 m
towers) and 500 MHz radar wind profiler. DH2 sorties consisted of one aircraft assigned to vertical profiling on 100
m diameter helix trajectories with 2 m s$^{-1}$ ascent/descent rates (launched about 5 min ahead of the others) to reconnoiter
stratified layer locations and depths, winds aloft, and identify turbulent layers. Other aircraft in the sortie were then
assigned to examine interesting layers more closely by either profiling the specific layer more slowly or more often,
or by conducting lateral surveys of these layers with elongated racetrack patterns up to 1.5 km long. Although the
aircraft had on-board lighting, manual control of launch and landing was extremely difficult in the dark, so automatic
control modes were used throughout the flights. A crew of four provided equipment setup, aircraft preparation, and
launch operations for each day's observations. Once the sortie was airborne, the crew supervised flight operations
from inside a surface vehicle, sending occasional commands to alter flight trajectories, with verbal communication
among the team to coordinate measurements, avoid high-wind altitudes, and plan for landing of the aircraft. Although
one person could have supervised the whole sortie, in principle, a multi-person operation reduced the workload and
resulted in improved communication with the science team. Sorties lasted approximately 75 minutes, and two sorties
were typically flown between 02:00 and 07:00 local time each day, ending well before any convective activity was
generated by insolation. A total of 72 DH2 flights were conducted in 31 multi-plane sorties, producing 106 hours of
measurements. Figure 6 shows one three-plane sortie with corresponding vertical profiles of high-resolution potential
temperature and turbulent kinetic energy dissipation rate, indicating a particularly turbulent layer between 2300 m and
2750 m AGL bounded by thin, strongly-stable sheets. Overview information and quick-look data plots from each
sortie are available at https://www.colorado.edu/p129765c7060/home/ideal_project and data from the campaign can
be accessed at https://www.eol.ucar.edu/field_projects/ideal.

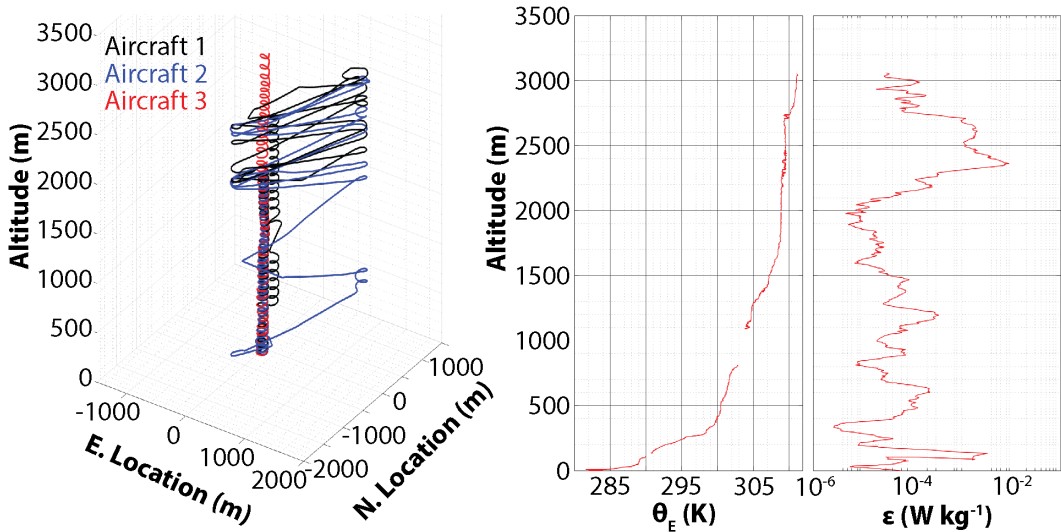

**Figure 6:** Representative DH2 flight trajectories from IDEAL (left), with high-vertical resolution postflight retrieval of virtual potential temperature (center) and turbulent kinetic energy dissipation rate (right).

Three campaigns, dubbed the Shigaraki UAV Radar Experiments (ShUREX) using DataHawk sUAS were conducted in the vicinity of Kyoto University's MU radar in Shigaraki, Japan, in June of 2015, 2016, and 2017. ShUREX 2015 used the original DataHawk vehicle with coldwire and Pitot turbulence sensors. DH2 were used in ShUREX 2016 and ShUREX 2017 carrying the pitot, as well as the new combined coldwire/hotwire turbulence instrument with the original (small diameter) protective shroud. All flights used a single DH2 vehicle launched from the ground at about

500 m MSL in altitude, reaching up to a maximum of 5 km MSL. Figure 7 shows a typical vertical profiling flight trajectory, along with high-vertical resolution profiles of potential temperature and the temperature structure constant $C_T^2$ that reveal thin layers of turbulence activity, often at the margins of well-mixed layers that are bounded by stable sheets as seen at 3000 m and 3300 m AGL. Objectives of these campaigns ranged from calibrating radar returns against in situ turbulence measurements, making radar-guided measurements of shear-driven Kelvin-Helmholtz

instabilities (KHI) in stratified layers, observing mid-level cloud-base turbulence (MCT), and quantifying turbulence growth in the convective boundary layer (CBL). These three campaigns yielded 86 DataHawk flights and over 112 hours of measurements, with analysis results reported in (Kantha et al., 2017; Kantha and Luce, 2018; Kantha et al., 2019; Luce et al., 2017; Luce et al., 2018a; Luce et al., 2018b; Luce et al., 2019).



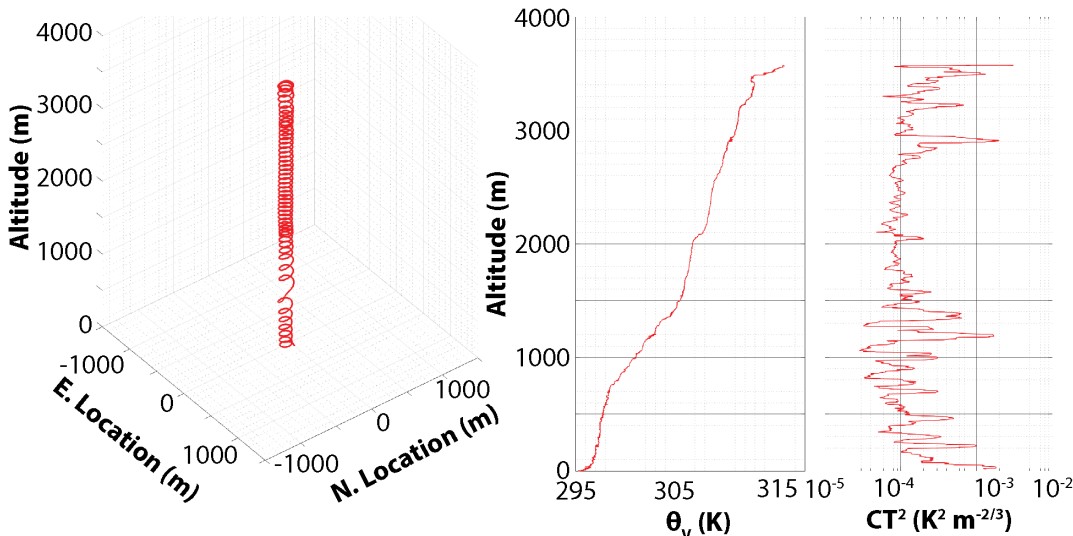


**Figure 7:** DH2 flight trajectory example from SHUREX 2017 (left), along with vertical profiles of virtual potential temperature (center) and temperature structure constant (right) derived from post-flight data processing.

In summer 2018, the DH2 was deployed to the San Luis Valley of Colorado as part of the Lower Atmospheric Profiling
at Elevation – a Remotely-Piloted Aircraft Team Experiment (LAPSE-RATE) campaign. During LAPSE-RATE, the
DH2 conducted repeated profiling of the lowest 500 m of the atmosphere over agricultural land in the western part of
the San Luis Valley. Doing so resulted in the sampling of a variety of conditions of interest, including the morning
transition from a stable to convective boundary layer, the daytime evolution of the convective boundary layer,
microscale circulations induced by the surrounding terrain, and outflow from convective storms that formed over the
mountains surrounding the San Luis Valley. In a series of studies (Jensen et al., 2021; Jensen et al., in prep), data from
the DH2 were used to support numerical experiments on the influence of assimilating UAS-based observations into a
high-resolution weather prediction system. These studies demonstrated that high-resolution profiling as conducted by
the DH2 significantly enhanced the model's ability to predict both local circulations (valley drainage flows) and
convective initiation and precipitation.


Most recently, in 2020 the DH2 was deployed to the central Arctic Ocean as part of the Multidisciplinary drifting
Observatory for the Study of Arctic Climate (MOSAiC, Shupe et al., 2021; de Boer et al., 2022b). For MOSAiC, the
DH2 had two primary sampling priorities: high-frequency profiling between the sea ice surface and 1 km altitude, and
horizontal flight to sample spatial variability resulting from surface heterogeneities. Because of the extreme northern
latitudes (up to the north pole) covered by the MOSAiC campaign, in preparation for this 6-month deployment, the
DH2 navigation system was updated with a differential GPS (DGPS) system to provide estimated azimuth angles.
This system was developed to replace azimuth angles provided by the magnetometer, since the magnetic field is nearly
vertical near the magnetic north pole, allowing the DH2 to be successfully operated at latitudes exceeding 87 N.
MOSAiC also saw significant weather challenges to operations, including cold temperatures (the aircraft was operated


down to -37 °C), low visibility and fog, high winds (the aircraft was operated in winds of 12 m s$^{-1}$), and a broken sea ice and melt-pond-covered surface environment. This last challenge is notable in that the small size and low cost of the DH2 allowed us to continue operating the aircraft despite very few dry areas for take-off and landing, resulting in a moderate risk for having the aircraft encounter melt pond water during those critical phases of flight. Despite these challenges, the DH2 conducted 82 flights during MOSAiC, resulting in 42.9 flight hours of data collected in this

unique location, and these data are contributing to publications focused on the lower polar atmosphere (e.g., Jozef et al., 2022; Dada et al., in prep.).

## 5 Summary and outlook

The DataHawk2 sUAS represents a novel observing platform for Earth system research. As discussed, the DH2 is a

custom system that has been configured to support detailed observing of the atmosphere, with a focus on thermodynamic, kinematic and turbulence properties. Deploying a customized suite of sensors, the DH2 has been deployed widely in support of a variety of atmospheric science focused field projects. These deployments have both contributed new perspectives on key atmospheric phenomena and supported the overall evolution and improvement of the system into its current form. To date, the DH2 has been operated by the University of Colorado and US

Department of Energy Atmospheric Radiation Measurement (ARM) program, although it is envisioned that other users will connect with the DH2 through collaborative research.

Looking forward, there are several additional system improvements planned. Some of the hardware implemented to support the recent addition of GPS-based navigation to the DH2 is already dated. Continued advancement and

miniaturization of GPS components makes it possible to integrate smaller and lighter DGPS units for navigation, and RTK (Real Time Kinematic) GPS components to improve system accuracy and support advanced navigation modes that would allow the platform to track a ground station. A need for such capabilities was brought to light during the MOSAiC campaign, when the aircraft positioning had to be updated constantly to adapt to the drifting sea ice floe. Additional platform improvements will target modification of the power system to support improved efficiency and

extended the flight endurance. This direction is also likely to benefit from continued advancement in battery technologies, and it is anticipated that DH2 flight times will continue to increase in the coming years beyond the current endurance of approximately one hour. From a sensing perspective, the current weak point of the system is its ability to make detailed, high-resolution wind measurements. While mean wind properties can be derived confidently for most flights, being able to measure the turbulent components of the wind would support enhanced abilities to

measure turbulent fluxes of heat and momentum, and quantities like turbulent kinetic energy. Such capabilities would extend the utility of the DH2 to better support research on turbulent flux structures throughout the lower atmosphere, wind energy-related research, and study of stable boundary layer conditions. To move toward this, planned advancements include improved measurement of the platform's true airspeed, which is currently impacted by airflow over the aircraft under certain flight maneuvers, and the potential integration of sensors to measure aircraft angle of

attack and sideslip.





In the coming years, continued deployment of the DH2 is envisioned, particularly to high-risk environments that require a small operational footprint and low-cost sensing system. Already, there are plans in place to take the DH2 to Antarctica to measure the atmospheric boundary layer there, and to continue collaborations with Japanese

colleagues interested in the turbulence of the lower atmosphere. As sUAS systems such as the DH2 continue to prove themselves in a variety of weather conditions and applications, it is expected that additional collaboration will develop with those who are interested in conducting atmospheric science research with small UAS. Additionally, current interest by operational weather forecasting entities, including the World Meteorological Organization, in the advancement of UAS to contribute to data collection in support of weather prediction could provide an expanded

opportunities for small, lightweight sUAS with well-characterized sensing capabilities to be regularly deployed around the world, providing detailed and frequent observations of the lower atmosphere that can be assimilated into operational weather forecasting activities.

## Data Availability

The MOSAiC DH2 data used in this manuscript are archived at the Arctic Data Center, see Jozef et al. (2021). The

MOSAiC radiosonde data are available at PANGAEA (Maturilli et al., 2021), and were produced via a partnership between the Alfred Wegener Institute, the DOE Atmospheric Radiation Measurement Program, and the German Weather Service. Where possible, we have cited other available DH2 data sets used in the example figures in this paper. Additional data not cited can be made available upon request to the corresponding author.

## Author Contribution

GdB and DL planned various DH2 data collection campaigns and acquired funding. DL led design, development and manufacturing of the DH2. JH and AD contributed to development, manufacturing, and testing of the DH2. DL, AD, JH and GdB acquired and analyzed DH2 data. JH led the preparation of the manuscript with contributions from all co-authors.

## Competing Interests

Co-author GdB has worked on a consulting basis for Black Swift Technologies, whose work is cited in the current manuscript.

## Acknowledgments

Development of the DataHawk2 sUAS has been supported by a variety of funding sources and people. Initial development of the DataHawk was supported by the US National Science Foundation (ITR-0427947, AGS-1041963)

and the Army Research Office (W911NF-12-2-0075). Initial testing of the DH2 payload was supported in part by the Cooperative Institute for Research in Environmental Sciences (CIRES) Innovative Research Program. Deployment of



the DH2 in Alaska was supported by the US Department of Energy Atmospheric System Research (ASR) program (DE-SC0011459 and DE-SC0013306), and Atmospheric Radiation Measurement (ARM) program funding. Support for the ShUREX and IDEAL campaigns was provided by the US National Science Foundation (AGS-1632829). The LAPSE-RATE campaign was supported in part by the US Department of Energy (DE-SC0018985) and US National Science Foundation (AGS-1807199). Support for MOSAiC operations was provided by the US National Science Foundation (OPP 1805569). Data used in this paper was produced as part of *RV Polarstern* (Knust, 2017) cruise AWI_PS122, the international Multidisciplinary drifting Observatory for the Study of the Arctic Climate (MOSAiC) with the tag MOSAiC20192020. We would like to thank the many people involved in supporting the MOSAiC expedition (Nixdorf et al., 2021). Gijs de Boer and Jonathan Hamilton were additionally supported by the NOAA Physical Sciences Laboratory.

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
