# Peer review of "The DataHawk2 Uncrewed Aircraft System for Atmospheric Research"

_Atmospheric Measurement Techniques, 2022_

## Referee Comment (RC1)

Referee Report on
**The DataHawk2 Uncrewed Aircraft System for Atmospheric Research**

The authors present an overview of their updated version of uncrewed aerial system (UAS) designed to simultaneously measure different thermodynamic properties of the atmosphere including turbulencs. The paper does a good job of reviewing the different components of the systems on the aircraft and provides evidence of validation of the successful operation of these systems.

I found the paper to be well written and, as noted, each system is generally well described. This information is potentially useful for other researchers interested in developing their own UAS, or for researchers who are interested in the data produced by the DataHawk2 who may wish to know more details about the systems which produced the data. I therefore recommend the article for publication in Atmospheric Measurement Techniques.

I did feel, however, that although the authors described the aircraft systems in great detail, they did not provide much information about the sensor data acquisition process, particularly given the asynchronous mix of what I expect would be both analog and digital data streams. Given that the role of the UAS is to essentially be a platform to carry the sensors, it is important that the system used to acquire and store the sensors' signals be described in more detail. For example, presumably the output from the fine-wire array is analog, what is the resolution of the ADC? Does it support serial communications? if so, which ones? How many sensors of different types can it support? How does it handle the asynchronous messages and ensure their timing? etc.

Other, minor comments that I also feel should be addressed before publication are as follows:

1. [Line 84] Although the authors use the more contemporary description of uncrewed aircraft systems, they still refer to traditional aircraft as manned aircraft. Perhaps rephrase to refer to crewed aircraft instead for consistency.

2. [footnote on page 2] I would also add unmanned/uncrewed aerial vehicles to the list of other designations used to describe UAS.

3. [Line 147] The DH2 acronym used for DataHawk2 is already defined on line 106. Also, once DH2 is defined, the authors occasionally seem to return to using the full name of DataHawk2 (e.g. lines 158 and 171)

4. [Section 2.3] Is their a reason that solar shielding has not been used for the RSS-421?

5. [Line 324] The authors mention a post-flight calibration process but do not provide the details until much later. Perhaps indicate to the reader that this process will be described in a later section.

6. [Line 382] As above, the wind comparison to the radiosonde is conducted prior to the description of how winds are obtained from the platform.

7. [Line 428] The authors mention their use of both ascent and descent portions of the profiles during calibration to cancel out lag-induced-offsets. This is the first mention of these offsets and no information is provided about their source. Are they due to sensor time response? Data acquisition system timing? More details are required.

8. [Line 458] The authors are using the pitot sensor for calibration of the hot-wire through spectral comparison. Frequency response of the pitot probe will play a role in this calibration and should be mentioned.

9. [Line 465] Which Kolmogorov constant is used by the authors in the inertial subrange model used to determine the dissipation rate? More details about this process would be beneficial.

10. [Figure 5] The authors present example spectrum to demonstrate the cold-wire measurement capability. A similar plot for the hot-wire should also be provided.

11. [Line 542] I would argue that the difference between the hybrid approach and radiosonde is as high as 2 m s−1 in the lowest 200 m of measurement. Do the authors have any insight as to why the disagreement of is approach with the traditional approach increases at lower altitude?

12. [Line 543] The authors mention confidence intervals for the wind estimate, but I could not find a description of how these intervals were determined.

13. [Line 563] The authors use the more-traditional capitalization of Pitot here, whereas in the rest of the manuscript the contemporary non-capitalized form is used. I believe AMT prefers the more contemporary form.

---

## Author Comment (AC1)

**Referee Comment Responses: The DataHawk2 Uncrewed Aircraft System for Atmospheric Research**

**Referee Comment #1: Sean Bailey**

*The authors present an overview of their updated version of uncrewed aerial system (UAS) designed to simultaneously measure different thermodynamic properties of the atmosphere including turbulencs. The paper does a good job of reviewing the different components of the systems on the aircraft and provides evidence of validation of the successful operation of these systems.*

*I found the paper to be well written and, as noted, each system is generally well described. This information is potentially useful for other researchers interested in developing their own UAS, or for researchers who are interested in the data produced by the DataHawk2 who may wish to know more details about the systems which produced the data. I therefore recommend the article for publication in Atmospheric Measurement Techniques.*

*I did feel, however, that although the authors described the aircraft systems in great detail, they did not provide much information about the sensor data acquisition process, particularly given the asynchronous mix of what I expect would be both analog and digital data streams. Given that the role of the UAS is to essentially be a platform to carry the sensors, it is important that the system used to acquire and store the sensors' signals be described in more detail. For example, presumably the output from the fine-wire array is analog, what is the resolution of the ADC? Does it support serial communications? if so, which ones? How many sensors of different types can it support? How does it handle the asynchronous messages and ensure their timing? etc.*

We would like to thank Dr. Bailey for the time put into this thorough and detailed review of the paper. To address some of the questions in the paragraph above, additional text about the overall sensor data handling process has been added (lines 266 – 276). Other sensor-specific details have been added in the Scientific Payload section.

*Other, minor comments that I also feel should be addressed before publication are as follows:*

1. *[Line 84] Although the authors use the more contemporary description of uncrewed aircraft systems, they still refer to traditional aircraft as manned aircraft. Perhaps rephrase to refer to crewed aircraft instead for consistency.*

The authors appreciate Dr. Bailey's attention to detail, have updated "manned" to "crewed" for consistency (revised draft line 86).

2. *[footnote on page 2] I would also add unmanned/uncrewed aerial vehicles to the list of other designations used to describe UAS.*

The authors agree that this would be a good addition and have added unmanned/uncrewed aerial vehicles to footnote as requested (footnote on page 2).

3. *[Line 147] The DH2 acronym used for DataHawk2 is already defined on line 106. Also, once DH2 is defined, the authors occasionally seem to return to using the full name of DataHawk2 (e.g. lines 158 and 171)*

The authors originally made an attempt to have the full aircraft name mentioned at the start of each section, but agree that this is confusing and appears inconsistent. We have changed all instances of DataHawk2 in the text body and figure captions to DH2 following the definition on revised draft line 108.

4. *[Section 2.3] Is their a reason that solar shielding has not been used for the RSS-421?*

Solar shielding has not been used on the RSS-421 in this case as it is not present in some Vaisala-designed applications with a very similar sensor package, such as the RS-41 radiosonde. The silver solar reflective coating on the temperature sensor helps mitigate solar effects in both our application and the RS-41 radiosonde. The authors have added a sentence (lines 313 – 314) to clarify this design choice for future readers: "The RSS-421 is unshielded

on the DH2, similar to the RS-41 application of these sensors; the silver solar reflective coating on the temperature sensor helps mitigate solar effects."

5. *[Line 324] The authors mention a post-flight calibration process but do not provide the details until much later. Perhaps indicate to the reader that this process will be described in a later section.*

The authors agree that adding this detail will help the readability of the paper, and have added "as detailed in Section 3" to revised draft line 345 for clarification.

6. *[Line 382] As above, the wind comparison to the radiosonde is conducted prior to the description of how winds are obtained from the platform.*

The authors agree that this is confusing and have decided to re-arrange the third section to create a more logical flow of information in the manuscript. Now, the wind comparison is conducted after the description of how the winds are obtained from the platform. Additionally, a reminder pointing the reader to the wind speed estimate calculations in Section 3.3 has been added to revised draft line 528.

7. *[Line 428] The authors mention their use of both ascent and descent portions of the profiles during calibration to cancel out lag-induced-offsets. This is the first mention of these offsets and no information is provided about their source. Are they due to sensor time response? Data acquisition system timing? More details are required.*

The authors agree that the cause of these lag-induced offsets should be clarified in the manuscript; these lag-induced offsets are due to sensor time response. The authors have added clarification about the source of the lag-induced offsets is added to lines 415 – 416, referring to the sensor time response differences discussed on line 413. The following phrase is added to the paper (lines 415 – 416): "… caused by these differing sensor time responses… ".

8. *[Line 458] The authors are using the pitot sensor for calibration of the hot-wire through spectral comparison. Frequency response of the pitot probe will play a role in this calibration and should be mentioned.*

The revision notes (lines 447 – 453) that the pitot frequency response is not a factor in the 800Hz measurements (400 Hz Nyquist frequency). Propeller vibration noise is the chief limitation, necessitating calibration of the hotwire at frequencies below 100 Hz.

9. *[Line 465] Which Kolmogorov constant is used by the authors in the inertial subrange model used to determine the dissipation rate? More details about this process would be beneficial.*

The details of the Kolmogorov model used are provided in the Frehlich 2003 reference cited in the original version. This citation is moved up in the revision (line 460) to make the source of the method clearer.

10. *[Figure 5] The authors present example spectrum to demonstrate the cold-wire measurement capability. A similar plot for the hot-wire should also be provided.*

The authors agree that a hotwire plot should also be provided and have updated Figure 5 (revised draft Figure 4) to include a hotwire plot.

11. *[Line 542] I would argue that the difference between the hybrid approach and radiosonde is as high as 2 m s−1 in the lowest 200 m of measurement. Do the authors have any insight as to why the disagreement of is approach with the traditional approach increases at lower altitude?*

The authors do not have a definitive reason for the disagreement in wind speed between the traditional and hybrid wind estimation approaches shown in the example flight (revised draft Figure 5). Fully determining the cause of this difference would require significant work that the authors feel falls outside the scope of the paper, but they can provide some insight. The hybrid approach is more heavily filtered,

which could explain some of the disagreement between the two estimation approaches, though it is unclear how this difference would affect the results seen in the example flight. Additionally, it should be noted that the disagreement between the DH2 and radiosonde's estimated wind speed at lower altitudes could be attributed to the temporal (up to an hour) and/or spatial difference (up to a kilometer) between the two flight trajectories.

We have changed "agree quite well" to "similar to one another" to better convey the differences between the wind estimation techniques (revised draft line 572).

12. *[Line 543] The authors mention confidence intervals for the wind estimate, but I could not find a description of how these intervals were determined.*

The authors have added a statement to line 576 "(detailed earlier in this section)" and changed "… calculated from" to "… calculated in" to clarify that these confidence intervals were described earlier and point the reader back to them if necessary. The re-organization of Section 3, placing the radiosonde comparison section at the end, puts the description of the confidence intervals closer to the wind estimate discussion, which the authors believe will help make the source of these calculations more clear. Additionally, some clarifications on the interval determination reasoning/methodology were made earlier in Section 3, shown below.

Replaced the following text:

"Therefore, a confidence interval was computed to determine a range for the actual difference between the radiosonde and DH2 sensors given the same 95 % significance level."

With a revised statement to clarify confidence interval methods:

"There is minimal usefulness in knowing that the two sensors are not absolutely the same; this is already assumed. However, knowing a range for the actual difference between the radiosonde and DH2 is of interest. Therefore, a confidence interval was computed to determine this actual difference between the sensors given the same 95 % significance level."

13. *[Line 563] The authors use the more-traditional capitalization of Pitot here, whereas in the rest of the manuscript the contemporary non-capitalized form is used. I believe AMT prefers the more contemporary form.*

The authors again thank Dr. Bailey for his attention to detail, and have corrected "Pitot" to the more contemporary "pitot" form.

---

## Author Comment (AC2)

**Referee Comment Responses: The DataHawk2 Uncrewed Aircraft System for Atmospheric Research**

**Referee Comment #2: Anonymous**

*The paper deals with describing the uncrewed aircraft system DataHaw2 capabilities for atmospheric studies. The focus is on atmospheric variables such as temperature, pressure and wind profiles, and ultimately provides data related with atmospheric turbulence. The topic is of great interest for atmospheric sciences and therefore suitable for publication in Atmospheric Measurement Techniques. The paper is well written and structured. A lot of details are given about the airplane, including instrument developments and deployments.*

*However, I have a major concern before recommending its publication in Atmospheric Measurement Techniques and is about the result sections: In its current form, many details are given about the flights performed in different campaigns, but there are no discussions of study cases that show the potential in the airplanes. Only a few graphs are given, but without discussions and even with no appropriate description of the variables represented. Such discussion must include results about atmospheric measurement/topics. Therefore, result section need to be improved*

We appreciate the reviewer's comments and thank the reviewer for taking the time to provide detailed feedback. Regarding this first concern, however, we believe that the reviewer has misinterpreted the vision for what is communicated in this article. Given the submittal to AMT, we have chosen to focus the article on the technology itself, and with that provide a brief overview of previous applications of that technology. The section that we believe the reviewer is referring to as the "results" section is intentionally labeled as "Previous Deployments and Scientific Use Cases" to provide a fair and accurate representation of what is in this section. This section was never intended to provide the results of detailed scientific analysis of observations collected during the campaigns that are offered as examples. Each such analysis would result in enough material to develop multiple individual additional publications, some of which have already been published and cited in this text. Therefore, we respectfully disagree with the reviewer's perspective that more "results" are required, as we are including this section to provide examples to the reader on how the technology described has previously been deployed. Additionally, with respect to the reviewer's comment about not having sufficient descriptions of the variables represented, we're not certain we understand this concern. The figure axes and captions clearly state which variables are presented, and the body of the text also specifically calls out which variables are shown and even how these measurements might be interpreted (e.g. discussion about how $CT^2$ is a proxy for turbulence in the atmosphere, etc.). We believe that we have provided an adequate level of detail for the reader to understand what is being presented, again keeping in mind that we are not attempting to provide a detailed analysis of these observations but are rather sharing these with the reader as an example of the types of measurements that the DataHawk2 can provide. Should the editorial team believe that we are misunderstanding the reviewers concern or that we are not adequately describing the variables shown in the figures, we would be happy to make modifications.

*My second point is not a concern. Indeed is a general comment that I would like the authors answer and if possible mention in the manuscript. Uncrewed aircrafts have a tremendous potential for atmospheric studies. However, there are many governmental limitations for flight operations, and that also varies with countries. Could the author provide their feedbacks about that and how to deal with it?*

The authors concur with the reviewer's comment that there are many governmental limitations on UAS operation that vary from country to country, vary with location in a country, and vary with time, and add that these limitations can sometimes have a substantial effect on the design of a flight campaign. The University of Colorado (where the DH2 is based) has a Director of Flight Operations (DFO) office that helps obtain flight permissions for the needs of a given campaign wherever CU aircraft are operated. CU has a blanket Certificate of Authorization that enables flight in eligible airspace across the USA up to 400 feet, and also has operators that have FAA Part 107 certification. Either of these avenues can be used for local flight testing and for flight campaign needs that don't require higher altitude flight (or other flight conditions not permissible under the COA/Part 107). When a certain flight campaign requires deviation from the COA/Part 107 rules in the USA, a specific COA is sought to

enable these different flight operations, for example, increasing the maximum flight altitude. In US restricted airspace, rules vary by location and by airspace management practices at that location. When CU operates internationally, the DFO helps UAS operators work with a foreign entity to obtain flight permissions. Many months are sometimes necessary to work out these case-by-case airspace permissions in advance of a campaign.

We have added a sentence to each campaign mentioned in the "Previous Deployments and Scientific Use Cases" section mentioning the flight rules under which each campaign was conducted (ERASMUS: lines 605 – 606, POPEYE/ICARUS: line 620, IDEAL: lines 646 – 647, ShUREX: 666 – 667, LAPSE-RATE: 683 – 684, MOSAiC: 701 – 703). We believe this will give the reader some insight into the complexities of UAS operations from a governmental permissions standpoint and provide potential permissions avenues for a group that may be planning a UAS flight campaign in the future.

*I also have some minor concerns that I believe must be addressed:*

*Introduction: I generally miss references in the Introduction section. For example, there are no references from line 30 to 40.*

We agree with the reviewer that this section could use some additional references and have added these into the text.

*Lines 60 – 61: I do not understand how the compositions of the atmosphere affect the uncertainties in remote sensing measurements*

We apologize for the misunderstanding – in this case "composition" was meant to encompass the presence of clouds and precipitation, which can attenuate the signal of many remote sensors. We have updated the text to use the phrase "properties", rather than "composition".

*Lines 62 – 63: Dial and Raman lidar for water vapor do not need particle backscattering. Please correct.*

The reviewer is correct in their statement that not all lidar systems require particles to provide a measurement. Our comment here was specific to lidar systems used for wind measurement and aerosol backscatter lidars that are used to derive information on the presence of particulates and hydrometeors. We have updated the text to reflect this.

*Lines 80 – 85: I would highlight the potential for studying spatial variability of atmospheric variables.*

We agree with the reviewer that highlighting the potential for studying spatial variability would be a good addition to this section. We have modified the last sentence in this paragraph (lines 86 – 88) to highlight the potential for studying spatial variability

Changed from: "Additionally, they provide greater horizontal resolution than tethered balloons, along with the ability to operate in higher wind conditions."

To: "Additionally, they provide enhanced perspectives on spatial variability compared to tethered balloons, along with the ability to operate in higher wind conditions."

*Line 93: After reading the manuscript, I did not find any instrument deployed in DataHaw2 for aerosol-cloud interactions. Is there a plan to install miniaturized instruments for that?*

We do not have plans to install a miniaturized aerosol instrument on the DH2 at this time, but this may be something we'd like to pursue in the future. The authors find aerosol-cloud interactions to be an interesting and valuable area of study that could make this endeavor worthwhile.

*Lines 160 – 170: I am confused. Is there a final version of DataHaw2 commercially available? What would be the final cost?*

The authors apologize for any confusion here; there is not a final version of the DH2 commercially available, and it would be very difficult to estimate the cost of the aircraft in a commercial setting, as this is highly dependent on labor cost and other factors. The authors would also like to note that there will likely never be a final version of the DH2, as the aircraft is designed to be adaptable to various research goals and will therefore be modified for each use case to some degree. The following sentence has been added to lines 173 – 175 for clarification: "The DH2 is not commercially available at this time, though the authors are open to future collaboration that would use the DH2 in its current configuration or a configuration evolved to meet the needs of a specific research project."

*Scientific Payload: I think that summarizing everything in a Table could make the paper easier to read.*

The authors agree that summarizing the scientific payload in a table makes the paper easier to read and have added a table at the start of the Scientific Payload section.

*Instrument Performance can be divided in several sub-sections to make the manuscript easier to read.*

The authors agree that adding sub-sections to the Sensor Performance and Evaluation section will make the manuscript easier to read. The following headings have been added.

- 3.1 Thermodynamic Properties
- 3.2 Turbulence Properties
- 3.3 Wind Estimation
- 3.4 Radiosonde Comparison: Example Flight from MOSAiC

To make the sections more logical in layout, the radiosonde comparison paragraphs/figures were moved to the end of the Sensor Performance and Evaluation section, and the IR temperature sensor paragraph was moved up to the new "Thermodynamic Properties" section.

*Figura 4 needs further explanation. I do not understand 'frequency' in Figure 4b*

The authors would like further clarification of this comment by the reviewer, if the editorial team feels it is appropriate. The plot in question (now labeled Figure 3 due to re-organization; was previously Figure 4) is a commonly used plot in spectral analysis and is described in the caption and discussed in lines 426 – 432.

*Section 4 ' Previous deployments and Scientific study cases': I think that a table summarizing all the campaigns and with the main flights characteristics and objectives could serve as a good illustration. Are data of the different campaigns free available? See also my main concern about the results section*

Respectfully, the authors believe they have provided sufficient detail for the overviewed campaigns and disagree that a table would add significant value to the paper. Further information on each campaign and the data available can be found in the paragraphs describing each campaign or the referenced works for each deployment (see lines 590 – 620 for ERASMUS, 622 – 647 for IDEAL, 653 – 667 for ShUREX, 673 – 684 for LAPSE-RATE, and 686 – 704 for MOSAiC). Additionally, the data used in the MOSAiC radiosonde comparison section is referenced in the "Data Availability" section of the paper. If the editorial team feels that the information provided is insufficient, the authors would be happy to make modifications to the current sections.

*Conclusions: I think this section need to be re-written. Authors focus more on negative points than in main achievements.*

The authors agree that more detail should be added on the achievements of the DH2, especially in the first paragraph (lines 706 – 716) of the "Summary and Outlook" section. Campaign locations have been added to the paragraph, along with a concluding sentence highlighting the total number of flight hours the DH2 has conducted, and mention of an upcoming Antarctic deployment.